# Oncogenic Role of ADAM32 in Hepatoblastoma: A Potential Molecular Target for Therapy

**DOI:** 10.3390/cancers14194732

**Published:** 2022-09-28

**Authors:** Takahiro Fukazawa, Keiji Tanimoto, Emi Yamaoka, Masato Kojima, Masami Kanawa, Nobuyuki Hirohashi, Eiso Hiyama

**Affiliations:** 1Natural Science Center for Basic Research and Development, Hiroshima University, Hiroshima 734-8553, Japan; 2Department of Radiation Disaster Medicine, Research Institute for Radiation Biology and Medicine, Hiroshima University, Hiroshima 734-8553, Japan; 3Department of Surgery, Graduate School of Biomedical and Health Sciences, Hiroshima University, Hiroshima 734-8553, Japan

**Keywords:** ADAM32, hepatoblastoma, stemness, cell mobility, apoptosis

## Abstract

**Simple Summary:**

Hepatoblastoma (HBL) is a rare hepatic malignancy occurring mainly in early childhood. Although recently developed treatment regimens have improved prognosis, many refractory cases still exist, and the anticancer drug cis-diamminedichloroplatinum—cisplatin—primarily used in HBL treatment often induces severe side effects. Therefore, more effective and safer therapies are needed. In this study, we demonstrate that the expression level of *ADAM32* is particularly high in HBL tissue samples and is associated with poor prognosis in various cancer types through the regulation of cancer cell proliferation, stemness, migration, invasion, and acquired resistance to chemotherapy. Our results thus indicate that ADAM32 could be a promising candidate therapeutic target molecule, and provide new insights into the molecular mechanisms of HBL carcinogenesis.

**Abstract:**

Outcomes of pediatric hepatoblastoma (HBL) have improved, but refractory cases still occur. More effective and safer drugs are needed that are based on molecular mechanisms. A disintegrin and metalloproteases (ADAMs) are expressed with high frequency in various human carcinomas and play an important role in cancer progression. In this study, we analyzed expression of *ADAMs* in HBL with a cDNA microarray dataset and found that the expression level of *ADAM32* is particularly high. To investigate the role of ADAM32 in cancer, forced expression or knockdown experiments were conducted with HepG2 and HBL primary cells. Colony formation, cell migration and invasion, and cell viability were increased in HepG2 expressing ADAM32, whereas knockdown of ADAM32 induced a decrease in these cellular functions. Quantitative RT-PCR demonstrated an association between *ADAM32* expression and the expression of genes related to cancer stem cells and epithelial–mesenchymal transition (EMT), suggesting a role of ADAM32 in cancer stemness and EMT. Furthermore, knockdown of ADAM32 increased cisplatin-induced apoptosis, and this effect was attenuated by a caspase-8 inhibitor, suggesting that ADAM32 plays a role in extrinsic apoptosis signaling. We conclude that ADAM32 plays a crucial role in progression of HBL, so it might be a promising molecular target in anticancer therapy.

## 1. Introduction

Hepatoblastoma (HBL) is a rare hepatic malignancy that occurs mainly in early childhood [1,2,3]. Although new treatment regimens for HBL have recently been developed and are improving prognosis with HBL, many refractory cases still exist [4,5]. Furthermore, the cis-diamminedichloroplatinum (CDDP) primarily used in HBL treatment often elicits severe side effects, such as ototoxicity, secondary leukemia, and cardiotoxicity [6,7,8]. Therefore, it is desirable to develop more effective and safer therapies for HBL. Recently, drugs that target molecular pathways have been developed for several types of cancer on the basis of knowledge of detailed molecular mechanisms [9,10]. Clinical trials targeting the Wnt/β-catenin signaling pathway are being performed at present for patients with leukemia and other cancers, but not for HBL, despite the reported role of this pathway in HBL [11,12,13,14]. Other genetic and epigenetic mechanisms in HBL are currently under investigation [15,16], and the molecular mechanisms of carcinogenesis and progression in HBL will need to be elucidated to facilitate the development of novel molecular targeting therapies.

A disintegrin and metalloproteases (ADAMs) have recently been investigated as novel targets for treating various types of cancer. Given that ADAMs are expressed at high levels in some types of carcinoma and have been shown to promote carcinogenesis and progression, ADAM inhibitors have been studied and found to be effective against cancers expressing ADAMs [17,18,19]. ADAMs were first discovered as sperm surface proteins and were thought to regulate fertilization [20,21,22]. However, some members of the ADAM family play roles in angiogenesis, neurogenesis, and development of the heart and neuromuscular junctions [23,24]. Furthermore, dysregulation of ADAMs promotes disease progression and cancer phenotypes [17,18,25]. 

The molecular mechanisms, expression, and function of ADAMs in HBL have not yet been sufficiently elucidated. Thus, we at first attempted to determine the expression profile of ADAMs in HBL tissues compared with that in noncancerous liver (NCL) by using a cDNA microarray dataset that includes data from patients with HBL, and found that ADAM32 was expressed predominantly in HBL tissue. One possible mechanism of oncogenic expression of ADAM32 that has been reported is a gain of chromosome 8 (where ADAM32 is located) in cases of HBL, but not in hepatocellular carcinoma (HCC) [26,27]. Furthermore, amplification of ADAM32 has been reported in a wide range of malignancies, including hepatic malignancy [28]. On the other hand, its biological function has not been clarified yet. We therefore studied the biological functions of ADAM32 in HBL with the aim of evaluating its potential as a target of anticancer therapy.

## 2. Materials and Methods

### 2.1. cDNA Microarray Analysis

A cDNA microarray dataset containing gene expression data related to HBL was obtained from the Gene Expression Omnibus. The dataset (GSE131329) was analyzed with GEO2R to identify differentially expressed genes (DEGs) for the ADAM and a disintegrin and metalloproteinase with thrombospondin motif (ADAMTS) families by comparing 14 paired tumor and NCL samples. DEGs were defined as those with fold change > 1.5 and multiple-testing (Benjamini–Hochberg false discovery rate) adjusted *p* values < 0.05.

### 2.2. Database Analysis

To observe the distribution of gene expression in normal tissues, RefEx was used as previously described [29]. Relative expression levels of *ADAM9* and *ADAM32* in various normal tissues were evaluated and compared by cap analysis gene expression (CAGE). All original data are available from RefEx or the Functional Annotation of the Mammalian Genome project (FANTOM).

### 2.3. Immunohistochemistry (IHC) 

HBL and normal liver specimens were fixed in 20% formalin, then embedded in paraffin. Sections of specimens were cut with thickness 4 μm, transferred to slides, and de-paraffinized. Subsequent IHC was performed using a catalyzed signal amplification kit (K1500, Dako, Santa Clara, CA, USA) according to the manufacturer’s protocol. Briefly, sections were permeabilized in 20 mM Tris-HCL (pH 9.0) at 120 °C for 10 min. Subsequently, endogenous biotin was blocked with Biotin Blocking System (X0590, Dako). Then, endogenous peroxidase was blocked with 3% hydrogen peroxide. The sections were incubated with a protein block solution and rabbit polyclonal anti-ADAM32 antibody (#PA5-106591, Invitrogen, Carlsbad, CA, USA; 1:1000) or polyclonal normal rabbit IgG (PM035, MBL, Nagoya, Japan) as a negative control diluted by Antibody Diluent with Background Reducing Components (S3022, Dako) at 4 °C overnight. After washing in Tris buffered saline with Tween (TBS-T), sections were incubated with CSA Rabbit Link (K1498, Dako) and subsequently with streptavidin–biotin complex followed by an amplification reagent. Then, sections were incubated with streptavidin–peroxidase. After washing with TBS-T, they were treated with DAB substrate (FUJIFILM Wako Pure Chemical Corporation, Osaka, Japan) for 5 min. After washing with distilled water, sections were counterstained with hematoxylin. Finally, sections were dehydrated and mounted with MOUNT-QUICK (DM01, Daido sangyo, Saitama, Japan). Specimen details are given in Appendix A. 

### 2.4. Plasmid Construction

For forced expression experiments, cDNA encoding for ADAM32 was amplified by PCR using Fast Start Taq DNA Polymerase (Roche, Basel, Switzerland). The amplified fragment was cloned into pcDNA3.1/V5-His (Invitrogen) and p3xFLAG-CMV™-10 Expression Vectors (Sigma-Aldrich, St. Louis, MO, USA). For the Tet-inducible experiment, the region encoding 3xFLAG-ADAM32 (aa20 to aa787) was subcloned into pRetroX-Tight-Pur (Clontech, Mountain View, CA, USA). 

For knockdown experiments, pSUPERIOR-puro (OligoEngine, Seattle, WA, USA) was used. Target sequences of shRNA were designed by using siDirect version 2 (http://sidirect2.rnai.jp, accessed on 27 July 2017) according to an established protocol [30]. The sense and antisense oligos for the target gene were annealed and subcloned into the pSUPERIOR-puro vector according to the manufacturer’s protocol. Appendix A shows the oligo sets and shRNA target sequences used. All constructs were confirmed by sequencing analysis. The shRNA vectors targeting the *ADAM32* gene and LacZ are referred to as shADAM32 and shLacZ, respectively. The vector expressing mutant type TP53 (R248W) was constructed in a previous study [31] and is referred to as pCMX-p53-R248W. 

### 2.5. Cell Culture

The HBL cell line HepG2 was obtained from the Japanese Cancer Research Resource Bank and the breast cancer cell line MCF7 was obtained from the American Type Culture Collection. Cells were seeded onto either dishes or plates (depending on the objective of the experiment) in RPMI containing 10% FBS and 100 μg/mL kanamycin (Sigma-Aldrich), and were maintained at 37 °C in 5% CO_2_. HepG2 cells were transfected with pVSV-G (Clontech) and pRetroX-Tight-Pur FLAG-ADAM32 sequentially. To obtain stable lines, cells were cultured in medium with G418 and puromycin. Stabilized cells are referred to as HepG2 Tet and were used for forced expression experiments in this study. At the time of seeding or 24 h later (depending on the experiment), HepG2 Tet cells were treated with doxycycline (Dox) to induce ADAM32 expression. We refer to the group treated with Dox as Group Dox (+) and the untreated group as Group Dox (−). Appendix A show the details of each experimental group and protocol. 

We refer to the groups transfected with shLacZ and shADAM32 as Control Group shLacZ and Group shADAM32, respectively. Appendix A show the detailed experimental groups and protocols. Stable cell lines expressing mutant type p53 were established in a previous study [31]. HepG2 cells transfected with pCMX-empty and pCMX-p53-R248W are referred to as HepG2 mock and HepG2 mt, respectively. For the inhibition of caspase-8 activity, Z-IETD-FMK (TONBO Biosciences, San Diego, CA, USA) was used.

### 2.6. Primary Cell Culture of HBL (HBCs)

HBL specimens of size 5 mm square resected from patients were further minced and seeded on a collagen type Ⅰ coated 6-well plate dish (4810-101, Iwaki, Tokyo, Japan) in Hepatocyte Medium (#5201, ScienCell Research Laboratories, Carlsbad, CA, USA), after which they were maintained at 37 °C in 5% CO_2_ for 7 days. Then, the cells were reseeded onto dishes for each experiment. For knockdown experiments, siRNA (Hs_ADAM32_7 FlexiTube siRNA; 1027417, Qiagen, Hilden, Germany) denoted as siADAM32, or AllStar Negative control siRNA (1027281, Qiagen) denoted as siN, was transfected into primary cells with Trans it X2 Reagent (MIR6000, Mirus, Madison, WI, USA). Appendix A shows the details of each experimental group and protocol. Details of the HBL case are summarized in Appendix A. 

### 2.7. Real-Time RT-PCR

Total RNA was extracted by using NucleoSpin^®^ RNA (Macherey-Nagel, Düren, Germany) according to the manufacturer’s protocol, and cDNA was synthesized with a High-Capacity cDNA Reverse Transcription Kit (Applied Biosystems, Foster City, CA, USA). Real-time RT-PCR was performed with a 7900HT (Applied Biosystems) and FastStart Universal Probe Master (Roche) by following the TaqMan probe methodology according to the manufacturer’s protocol. *ACTB* (4326315E, Applied Biosystems) was used as the internal control. Appendix A shows the primer and probe sets.

### 2.8. Immunoblotting

Whole-cell extracts from cell pellets were prepared as previously described [32]. A total of 20 or 40 μg of each sample was resolved by using 5–12% gradient sodium dodecyl sulfate-polyacrylamide gel electrophoresis (ATTO, Tokyo, Japan) as appropriate and was transferred onto polyvinylidene membranes (Millipore, Burlington, MA, USA). To block nonspecific antibody binding, membranes were incubated with 2% bovine serum albumin or 5% skimmed milk in tris-buffered saline (TBS) for 1 h at room temperature. Subsequently, membranes were incubated with primary antibodies diluted in Can Get signal^®^ primary buffer (Toyobo, Osaka, Japan) overnight at 4 °C. After being washed with TBST, membranes were incubated with horseradish peroxidase (HRP)-linked antirabbit IgG (NA934U; GE Healthcare, Arlington Heights, IL, USA) or antimouse IgG (NA931V; GE Healthcare) diluted in a Can Get signal secondary antibody buffer for 1 h at room temperature. After being washed with TBST, membranes were incubated with Pierce Western Blotting Substrate Femto (Pierce, Rockford, IL, USA) and developed on an X-ray film (GE Healthcare). Appendix A describes the antibodies and their dilution conditions. All the whole immunoblotting figures can be found in Appendix A.

### 2.9. MTT Assay

Cell viability was evaluated with the MTT assay as previously described [33]. Cells were seeded at 5 or 1.5 × 10^3^ cells per well onto a 96-well plate and treated with Dox or transfected with shRNA vectors or siRNA (Appendix A). To analyze the effects of ADAM32 forced expression or knockdown on cell viability, cells were treated with CDDP (Appendix A). At the indicated time points, 0.4% MTT solution and 0.1 M monosodium succinate were added sequentially. The MTT formazan precipitate was dissolved in DMSO. Absorbance at 570 and 650 nm was measured.

### 2.10. Colony Formation Assay

Appendix A show details of each experimental procedure. For the ADAM32 forced experiment, HepG2 Tet cells were seeded onto six-well plates at 5 × 10^3^ per well and maintained in culture media with or without Dox for two weeks. For the knockdown experiments, native HepG2 and MCF7 cells transfected with shLacZ or shADAM32 were seeded onto six-well plates and cultured in media containing puromycin at 1 μg/mL. HBCs transfected with siN or siADAM32 were seeded onto six-well plates and cultured. Colonies that had formed after ten days or two weeks were stained with 0.5% crystal violet in 25% methanol. The number of colonies was counted as previously described [34,35].

### 2.11. Wound-Healing Assay

To evaluate cell migration, the wound-healing assay was performed by using Culture-Insert (ibidi GmbH, Gräfelfing, Germany). Appendix A show the details of each experiment. HepG2 Tet, HepG2, HBCs, and MCF7 cells were seeded into each culture insert. Thereafter, HepG2 Tet cells were treated with Dox for 48 h. On the other hand, HepG2, HBCs, and MCF7 cells were transfected with shRNA vectors or siRNA. After 24 h of incubation, the culture inserts were gently removed and further incubated with culture media. To evaluate wound healing and closure, phase-contrast images were captured at 18, 24, 36, 48, and 72 h, and percent wound-healing values were quantified and calculated with ImageJ (NIH) as in previous studies [36].

### 2.12. Cell Invasion Assay

The cell invasion assay was performed with a BioCoat^TM^ Matrigel^®^ Invasion Chamber (Corning, Corning, NY, USA). First, HepG2 Tet, HepG2, HBCs, or MCF7 cells were treated with Dox or transfected with shRNA vectors or siRNA for the forced expression or knockdown of ADAM32. Treated cells were seeded at 1 or 5 × 10^4^ into the upper chambers in culture media without serum. Thereafter, the lower chambers were filled with culture media containing 10% FBS as a chemoattractant. The chambers were maintained in this state for 24 or 48 h, after which residual cells on the top of the Transwell membrane were wiped off with a wet cotton swab. Cells that migrated into the Transwell were fixed with 4% paraformaldehyde and were stained with 0.3% methylene blue, after which they were counted under a light microscope (Appendix A).

### 2.13. TUNEL Assay

The TUNEL assay was performed to evaluate apoptosis by using the Apoptosis in situ Detection Kit (FUJIFILM Wako Pure Chemical Corporation). Cells were treated with Dox or transfected with shRNA vectors or siRNA, after which they were either treated with CDDP or not treated, and after 72 h they were harvested (Appendix A). Cells were then suspended in 4% paraformaldehyde, dropped onto glass slides, and dried at room temperature. TUNEL staining was performed according to the manufacturer’s protocol. After staining, phase-contrast images were captured and the number of TUNEL-stained cells was counted in five independent fields. The percentage of TUNEL-positive cells was then calculated.

### 2.14. Statistical Analysis

All statistical analyses except for the identification of DEGs in the microarray analysis were conducted with SPSS Statistics version 17.0 (IBM); DEGs in the microarray dataset were analyzed in GEO2R. Adjusted *p* values were calculated by applying the Benjamini–Hochberg FDR procedure. For pairwise comparisons between experimental groups, *t*-tests were performed. For comparisons among three or more experimental groups, one-way ANOVA was performed, and Tukey’s test was conducted for post hoc pairwise comparisons. *p* values < 0.05 were considered statistically significant. 

## 3. Results

### 3.1. ADAM32 Expressed in HBL and Associated with Clinical Course of Some Cancers

As ADAM family members have recently been demonstrated to be associated with cancer progression, and their inhibitors have been developed as anticancer drugs [17,18], we at first focused on the biological features of ADAMs in HBL. In the cDNA microarray dataset analyzed with GEO2R, expression levels of *ADAM32*, *ADAMTS6*, and *ADAM9* in HBL tumor tissues were 1.5-fold higher than in NCL. By contrast, expression of each of *ADAMTS1*, *ADAM19*, *ADAMTS13*, and *ADAMTS17* in tumor tissue was 0.75-fold lower than that in NCL. The other members of ADAMs and ADAMTSs were unchanged (Table 1). Thereafter, we analyzed the expression levels of *ADAM9* and *ADAM32* in various normal tissues by using the gene expression database RefEx to clarify the distribution of expression. We found that *ADAM32* expression was limited to the testis and reproductive system, whereas *ADAM9* was ubiquitously expressed in the adult body (Appendix A). Among ADAM family members, *ADAM32* expression was high in HBL but limited in normal tissues. These results were further confirmed by IHC with clinical samples. Level of expression of ADAM32 was high in embryonal and fetal subtypes of HBL, but weak in normal liver tissue and HBL macrotrabecular subtype, a rare subtype accounting for around 2% of total HBL that has pathlogical features similar to those of HCC (Figure 1) [37,38]. Thus, we attempted to clarify the biological function of ADAM32 in HBL (below). Database analyses performed to reveal the prognostic role of *ADAM32* in other types of cancer (by using the Kaplan–Meier plot) showed that higher expression of *ADAM32* is correlated with poorer prognosis in breast, ovarian, lung, and gastric cancers, whereas it is correlated with better prognosis in HCC (Appendix A).

### 3.2. ADAM32 Increased Viability of HBL

To clarify the biological functions of ADAM32, we performed functional analyses using forced expression and knockdown experiments in the HBL cell line HepG2 and primary cells of HBL (HBCs). In experiments of forced expression, the tetracycline induction system was employed to establish HepG2 cells with a controllable expression of ADAM32. Thereafter, we confirmed that HepG2 expressed ADAM32 and FLAG for 96–144 h after Dox induction (Figure 2a). For knockdown experiments, transient transfection with shRNA specific to *ADAM32* was able to reduce expression levels in HepG2 within 48 h. Transfection with siADAM32 in HBCs also reduced expression level at 48 h after transfection. Given that the sh#2 of shADAM32s decreased ADAM32 expression most effectively, we used sh#2 for all subsequent knockdown experiments (Figure 2b). 

When ADAM32 expression was forced, cell viability of HepG2 Tet in Group Dox (+) increased at 120 h (Figure 2c). Conversely, the knockdown experiment showed that cell viability of HepG2 in Group shADAM32 was significantly lower at 72 h than in Group shLacZ (Figure 2d). Cell viability of HBCs in Group siADAM32 was significantly lower at 120 h than in Group siN (Figure 2d). Moreover, the number of colonies increased in the colony formation experiment when expression of ADAM32 was forced (Figure 2e,f). By contrast, the number of colonies was reduced in Group shADAM32 of HepG2 and Group siADAM32 of HBCs (Figure 2g–j). Similar oncogenic functions of ADAM32 in HBL and HBCs were observed in experiments using breast cancer cell line MCF7 (Appendix A).

### 3.3. ADAM32 Regulates Expression of Stemness-Related Genes

Real-time RT-PCR demonstrated that expression levels of *ADAM32* were increased in Group Dox (+). Expression levels of stem cell-related genes *PROM1* and *ALDH1A1* were increased in Group Dox (+), whereas expression of *POU5F1*, *SOX2*, *KLF4*, *NANOG*, and *ABCG2* remained unchanged. Expression of *CD44* could not be detected. Expression levels of the Wnt/β-catenin target genes *MYC* and *CCND1* did not change (Figure 3a). Among stem cell-related genes, expression levels of *ADAM32* and *KLF4* decreased in Group shADAM32 of the HepG2 cells (Figure 3b), and expression levels of *ADAM32* and *ALDH1A1* decreased in Group siADAM32 in HBCs (Figure 3c). Furthermore, the expression levels of *ADAM32*, *POU5F1*, and *CD44* decreased in Group shADAM32 in MCF7 cells (Appendix A).

### 3.4. ADAM32 Regulates Cell Migration and Invasion

To evaluate the role of ADAM32 in cell migration and invasion, a wound-healing assay and a Matrigel invasion assay were performed. In the wound-healing assay, the ratio of wound healing increased in both Group Dox (−) and Group Dox (+), but was higher in Group Dox (+) at 48 h and 72 h (Figure 4a,b). In the knockdown experiments, the ratios of wound healing in Group shADAM32 or siADAM32 of HepG2 and HBCs were lower than those in Group shLacZ or siN (Figure 4c–f). Cell invasion activity was higher in Group Dox (+) (Figure 4g,h). In contrast, cell invasion activity was lower in Group shADAM32 or siADAM32 in HepG2 and HBCs (Figure 4i–l). Cellular functions of ADAM32 observed in HBL and HBCs were also confirmed in experiments using MCF7 cells (Appendix A).

Real-time RT-PCR showed that the expression levels of EMT-related genes *TWIST* and *VIM* increased in Group Dox (+), but expression of *HMGA2*, *SNAI1*, *SNAI2*, *CDH1*, and *CDH2* remained unchanged (Figure 4m). In ADAM32 knockdown experiments, expression levels of *SNAI2* decreased in Group shADAM32 in HepG2 cells (Figure 4n). Expression levels of *CDH2* decreased in Group siADAM32 in HBCs (Figure 4o). Immunoblotting showed that expression levels of E-cadherin and N-cadherin remained unchanged in HepG2 Tet, HepG2, and HBCs (Figure 4p). Expression levels of *HMGA2* were decreased in Group shADAM32 in the MCF7 cell line (Appendix A).

### 3.5. ADAM32 Promotes Antiapoptotic Signal

To understand whether ADAM32 could play a role in anticancer treatment, we conducted the MTT assay and TUNEL assay in the ADAM32 forced or knockdown cell lines. The appropriate concentration of CDDP in cell lines was first determined by observing activation of the apoptotic signal through p53 and caspase-3 for each cell type (Appendix A). The MTT assay showed that cell viability in HepG2 Tet cells after CDDP treatment was higher at 0 and 500 ng/mL in Group Dox (+), whereas values were similar at 2000 and 1000 ng/mL (Figure 5a). The TUNEL assay also showed that percentages of TUNEL-positive cells did not differ between the groups despite addition of CDDP at 2000 ng/mL (Figure 5b,c). However, cell viability of both HepG2 and HBCs treated with CDDP were lower in Group shADAM32 and Group siADAM32 at various concentrations (Figure 5f,k). Furthermore, the TUNEL assay showed that percentages of TUNEL-positive cells were higher in Group shADAM32 and Group siADAM32 of both HepG2 and HBCs treated with 500 or 2000 ng/mL CDDP, respectively (Figure 5g,h,l,m). Antiapoptotic functions of ADAM32 were also observed in MCF7, although these seemed to be weaker than those in HepG2 (Appendix A).

We then evaluated expression levels of apoptosis-related proteins and genes. Immunoblotting and real-time RT-PCR demonstrated similarity between Group Dox (−) and Group Dox (+) in phospho-ATM, total-ATM, phospho-p53, p53, cleaved caspase-3, and cleaved caspase-8 protein expression levels, as well as *BAX*/*BCL2* ratio, in the HepG2 cell line treated with CDDP (Figure 5d,e). On the other hand, immunoblotting demonstrated that expression levels of cleaved caspase-3 and cleaved caspase-8 were higher in Group shADAM32 of HepG2 treated with CDDP, while expression levels of phospho-ATM, total-ATM, phospho-p53, and total-p53, however, did not differ between Groups shLacZ and shADAM32 of HepG2, and Groups siN and siADAM32 of HBCs (Figure 5i,n). Real-time RT-PCR did not show a difference in *BAX*/*BCL2* ratios (Figure 5j,o). Protein and gene expressions in Group shLacZ and Group shADAM32 of MCF7 were similar to those of HepG2, although their ratios seemed to be lower in MCF7 than in HepG2 (Appendix A).

### 3.6. ADAM32 Regulates Antiapoptotic Functions through Caspase-8 

To clarify the mechanism of increased apoptosis when ADAM32 expression is knocked down, we focused on p53 by using the HepG2 mt cell line where the downstream signal is inhibited by constitutive expression of the mutant type of p53 [31]. Expression levels of the typical target genes *BAX* and *CDKN1A* were dramatically decreased in the HepG2 mt cell line, indicating that p53 downstream signaling was inhibited (Appendix A). The MTT assay showed that cell viability of HepG2 mt treated with CDDP was lower than that of the mock control. However, ADAM32 knockdown similarly affected cell viability of both HepG2 mock and mt treated with CDDP (Figure 6a). Thereafter, the TUNEL assay demonstrated that ADAM32 knockdown increased the percentage of TUNEL-positive cells similarly in both HepG2 mock and mt treated with CDDP but more notably so in the HepG2 mt cell line (Figure 6b,c). Immunoblotting demonstrated that the expression levels of phospho-ATM, phospho-p53, and total-p53 increased with CDDP treatment in both HepG2 mock and mt, and were higher in HepG2 mt. No differences in expression levels of these proteins were observed between Groups shLacZ and shADAM32. By contrast, expression levels of cleaved caspase-3 and cleaved caspase-8 in Group shADAM32 increased in both HepG2 mock and mt. However, the increase was more pronounced in HepG2 mt (Figure 6d). The *BAX*/*BCL2* ratios were increased with CDDP treatment in HepG2 mock but not in HepG2 mt. These ratios did not differ between Group shLacZ and Group shADAM32 (Figure 6e). 

To clarify the involvement of caspase-8, one of the critical factors in the extrinsic apoptosis signaling pathways in CDDP-induced apoptosis, we performed MTT and TUNEL assays by using the caspase-8 inhibitor Z-IETD-FMK. The MTT assay showed that cell viability after CDDP treatment was lower in Group shADAM32 treated with DMSO and 30 μM Z-IETD-FMK but not with 50 μM Z-IETD-FMK (Figure 6f). The TUNEL assay demonstrated that the increase in percentage of TUNEL-positive cells in Group shADAM32 was attenuated with treatment by Z-IETD-FMK (50 μM) (Figure 6g,h). Further analyses revealed that the induction of cleaved caspase-3 in Group shADAM32 treated with CDDP was diminished by treatment with Z-IETD-FMK, whereas expression levels of phospho-ATM, total-ATM, phospho-p53, total-p53, and cleaved caspase-8 were not affected (Figure 6i).

## 4. Discussion

Considering that the standard treatment for HBL is not successful for all patients, more effective therapeutic approaches, such as molecular targeting therapy, are needed [4,5]. In the present study, we first attempted to identify by microarray analysis which ADAMs are differentially expressed in HBL; we found that *ADAM32*, *ADAM9*, and *ADAM19* demonstrated noteworthy expression in HBL. Furthermore, the distribution of these genes in normal tissues evaluated by using a database analysis revealed that *ADAM32* expression was limited to the testis and reproductive systems in normal tissues. By contrast, *ADAM9* was ubiquitously expressed in the adult body. Furthermore, higher expression of ADAM32 was observed in HBL but not in normal liver tissue by using IHC. Given these observations, we clarified the biological function of ADAM32 considering that it could be a valid molecular target for anticancer therapy.

Forced expression and knockdown experiments directed at ADAM32 in the HBL cell line HepG2, primary cells of HBL (HBCs), and the breast cancer cell line MCF7 were performed to clarify its biological functions because prognostic importance of *ADAM32* has been observed in breast cancer (Appendix A). First, evaluation of cell viability with the MTT assay demonstrated that ADAM32 could increase cell viability in the HBL, HBCs, and breast cancer cell lines evaluated by forced expression and knockdown experiments. The colony formation assay further demonstrated that ADAM32 was able to promote colony formation. These results indicate that ADAM32 has functions related to cell proliferation and colony formation. Therefore, ADAM32 may play a role in cancer stem cell phenotypes. Expression levels of the stem cell-related genes fluctuated after modification of ADAM32 expression, supporting the conclusion that ADAM32 plays a role in promotion of cell stemness. Although an aberrant Wnt/β-catenin signal is important in HBL and has a role in cancer stemness [12,39,40,41], expression of the downstream target genes *MYC* and *CCND1* did not change with modified ADAM32 expression, suggesting that regulation of stemness by ADAM32 might be independent of the Wnt/β-catenin signaling pathway. 

We further investigated migration, invasion capacity, and related gene expression and found that ADAM32 also increased the migration and invasion ability of cells evaluated by forced and knockdown experiments. Expression levels of the EMT marker genes also fluctuated after modification of ADAM32 expression, suggesting that ADAM32 might play a role in EMT regulation. However, not all EMT marker genes evaluated in this study were affected by ADAM32, suggesting that other mechanisms are present in ADAM32-induced migration and invasion. Therefore, further investigation is needed to clarify the mechanisms by which ADAM32 regulates these cellular functions.

Some of the ADAM family members are known to be associated with resistance to anticancer therapy, which results in poor prognosis [17,18]. We therefore evaluated the effect of ADAM32 expression on apoptosis induced by anticancer drugs. CDDP is commonly used as a standard clinical treatment for various types of cancer, including HBL [42]. Knockdown of ADAM32 increased CDDP-induced TUNEL-positive apoptotic cells, whereas its forced expression did not affect the cells, suggesting that either a certain degree of ADAM32 expression might be required for its antiapoptotic function or ADAM32 might form a component part of a larger antiapoptotic machinery. In the future, it would be useful to evaluate the effect of ADAM32 on other anticancer treatments, such as other conventional anticancer chemotherapies, radiotherapies, or immunotherapies.

We further attempted to investigate the detailed molecular mechanisms underlying the antiapoptotic function of ADAM32 against anticancer drugs. Apoptosis is regulated by both intrinsic and extrinsic signals [43]. In the present study, levels of the apoptosis executing proteins cleaved caspase-3 and cleaved caspase-8, which are constituents of the extrinsic apoptosis signaling pathway, increased with CDDP treatment in ADAM32 knockdown cells. However, the activity of the intrinsic apoptosis signal-related proteins ATM and p53, and *BAX*/*BCL2* ratios, were not altered. These results suggest that the antiapoptotic function of ADAM32 occurs mainly via extrinsic apoptosis pathways. This was further supported by the observation that ADAM32 knockdown induced apoptosis in the HepG2 cells expressing the mutant type of p53, whereas intrinsic apoptosis signals were inhibited. Thus, we focused on extrinsic signals regulated by the FAS/caspase-8 pathway [44] and found that the caspase-8 inhibitor Z-IETD-FMK attenuated ADAM32 knockdown-induced apoptosis. This suggests an important role of the caspase-8 pathway in the antiapoptotic function of ADAM32. It was demonstrated that shedding of the FAS ligand prevented apoptosis and that some ADAM family members can induce shedding of the FAS ligand, resulting in modulation of its signal [45,46]. Therefore, ADAM32 might interact with the FAS/caspase-8-related protein and modulate apoptosis. This pathway will need to be further investigated in the future. The present study also suggests that status of caspase-3 might be important in ADAM32 knockdown-induced apoptosis. The levels of cleaved caspase-3 in MCF7 were much lower than those in HepG2, resulting in greater resistance to CDDP treatment. This suggests that ADAM32 in HBL cells performs a wide range of functional roles in apoptotic signaling, although our study is to date limited to in vitro investigations.

## 5. Conclusions

We demonstrated for the first time that ADAM32 plays a crucial role in HBL via regulation of cancer cell proliferation, stemness, migration, invasion, and acquired resistance to therapy via an extrinsic apoptotic signal. Our results indicate that ADAM32 could be a promising candidate for therapy targeting molecular pathways, although further studies will be needed to better illuminate the detailed mechanisms of action of ADAM32, both in vitro and in vivo, to evaluate its applicability in the clinic.

## Figures and Tables

**Figure 1 cancers-14-04732-f001:**
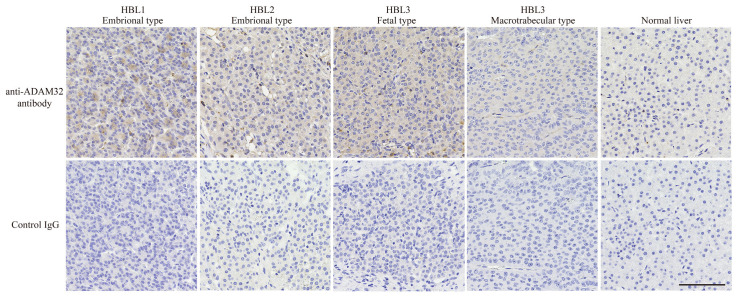
ADAM32 was expressed at high levels in embryonal and fetal subtypes of HBL. Expression of ADAM32 was evaluated with IHC using anti-ADAM32 antibody (**top**), and normal rabbit IgG as control (**bottom**) in the section of HBL subtypes. Results with normal liver tissue are shown on the far right. Representative images are shown. Scale bar = 100 μm.

**Figure 2 cancers-14-04732-f002:**
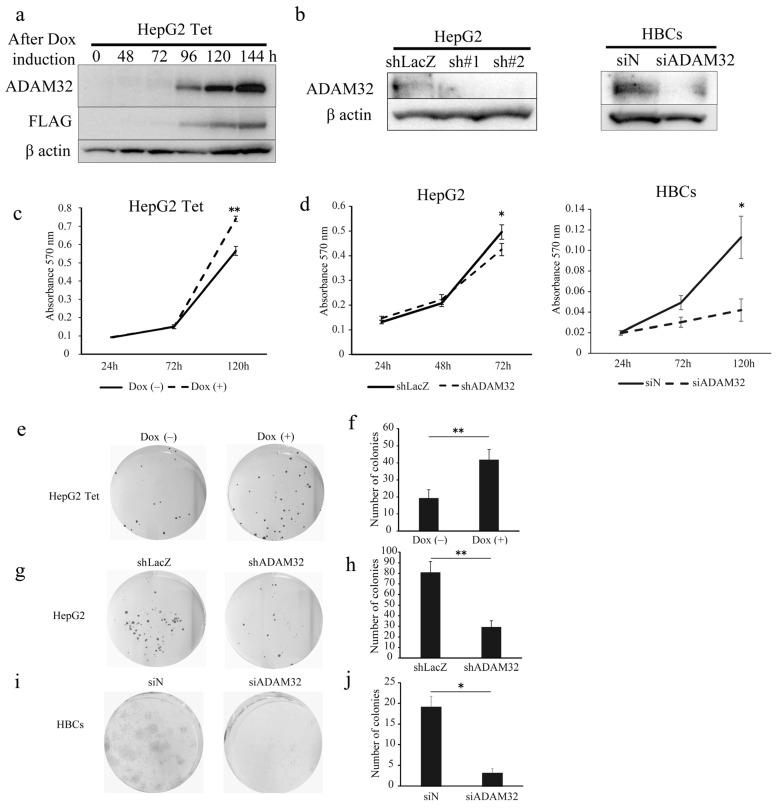
ADAM32 increases cell viability and colony formation in HBL and HBCs. (**a**) Immunoblotting detection with anti-ADAM32 (**top**), anti-FLAG (**middle**), and anti-β actin (**bottom**) was conducted on whole-cell extracts prepared from HepG2 Tet at each time point. Representative images from three independent experiments are shown. (**b**) Immunoblotting showing expression levels of ADAM32 in HepG2 cells (**left**) and HBCs (**right**) transiently transfected with shRNA or siRNA. (**c**) Cell viability of HepG2 Tet cells (n = 3). (**d**) HepG2 cells (**left**) and HBCs (**right**) evaluated in the MTT assay (n = 3). Representative images of (**e**) HepG2 Tet, (**g**) HepG2, and (**i**) HBCs in the colony formation experiment. The number of colonies was counted, and the averages (and SE) are shown for (**f**) HepG2 Tet (n = 6), (**h**) HepG2 (n = 6), and (**j**) HBCs (n = 3). * *p* < 0.05; ** *p* < 0.01. All the whole immunoblotting figures can be found in Appendix A.

**Figure 3 cancers-14-04732-f003:**
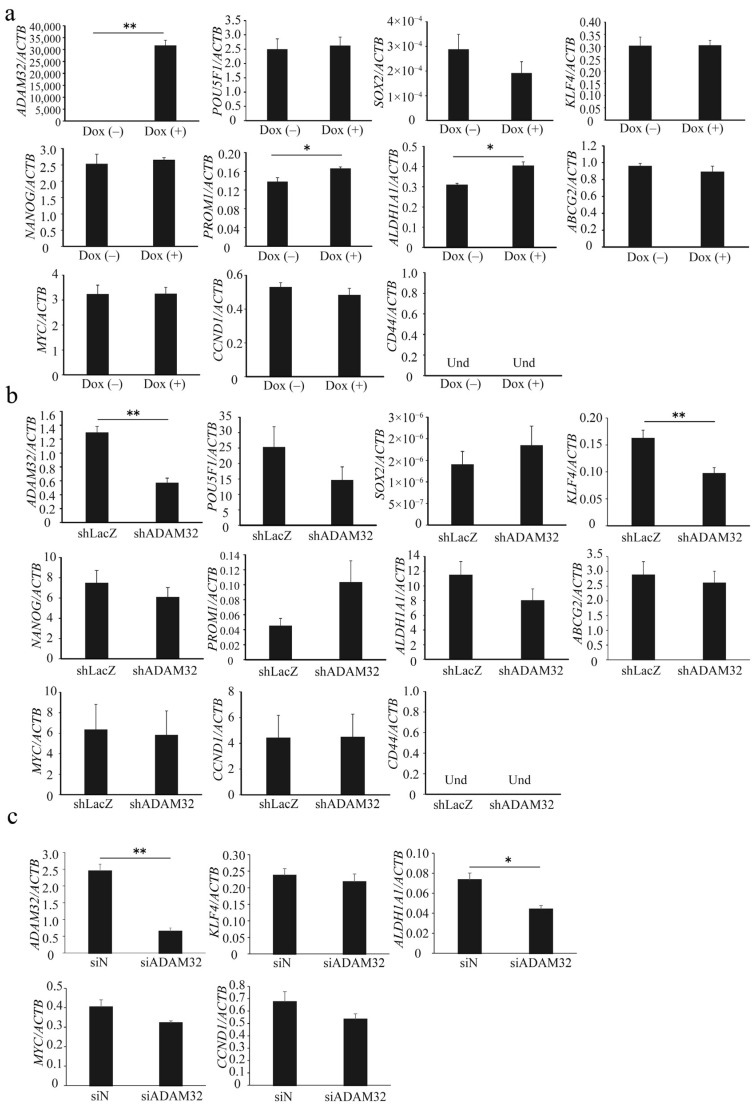
ADAM32 increases expression levels of stem cell-related genes. The expression levels were evaluated with real-time RT-PCR by using total RNA prepared from (**a**) HepG2 Tet treated without or with Dox, (**b**) HepG2, and (**c**) HBCs transiently transfected with shRNA or siRNA and incubated for 48 h. Relative gene expression levels were calculated as the ratio relative to *ACTB* levels. Values are summarized by mean and SE (n = 3); * *p* < 0.05; ** *p* < 0.01; Und, undetermined.

**Figure 4 cancers-14-04732-f004:**
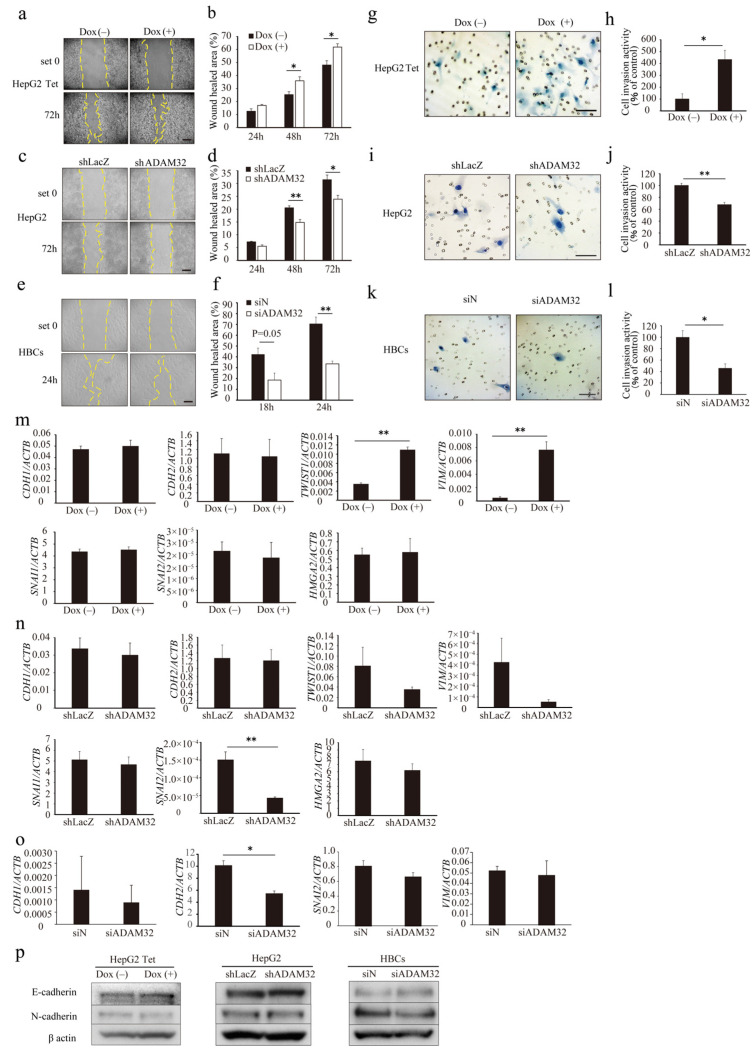
ADAM32 induces cell migration and invasion. Representative images for (**a**) HepG2 Tet, (**c**) HepG2, and (**e**) HBCs transiently transfected with shRNA or siRNA and incubated for 48 h in the wound-healing assays. Calculated proportions of healed areas (%) are shown in bar graphs for (**b**) HepG2 Tet (n = 4), (**d**) HepG2 (n = 4), and (**f**) HBCs (n = 3). Representative images from the cell invasion assay are shown for (**g**) HepG2 Tet, (**i**) HepG2, and (**k**) HBCs transiently transfected with shRNA or siRNA and incubated for 48 h. The calculated ratios (%) of invading cell number to control are shown in bar graphs for (**h**) HepG2 Tet (n = 4), (**j**) HepG2 (n = 3), and (**l**) HBCs (n = 3). Expression levels of EMT-related genes were evaluated with real-time RT-PCR by using total RNA prepared from (**m**) HepG2 Tet with or without Dox for 96 h, (**n**) HepG2, and (**o**) HBCs transiently transfected with shRNA or siRNA and incubated for 48 h. Relative gene expression levels were calculated as the ratio to *ACTB* levels (n = 3). (**p**) Expression levels of EMT-related proteins were evaluated with immunoblotting using protein prepared from HepG2 Tet, with or without Dox for 96 h, and HepG2 and HBCs transiently transfected with shRNA or siRNA and incubated for 48 h. Scale bar = 200 μm. (**b**,**d**,**f**,**h**,**j**,**l**–**o**) Values are summarized by mean and SE. * *p* < 0.05; ** *p* < 0.01; Und, undetermined. All the whole immunoblotting figures can be found in Appendix A.

**Figure 5 cancers-14-04732-f005:**
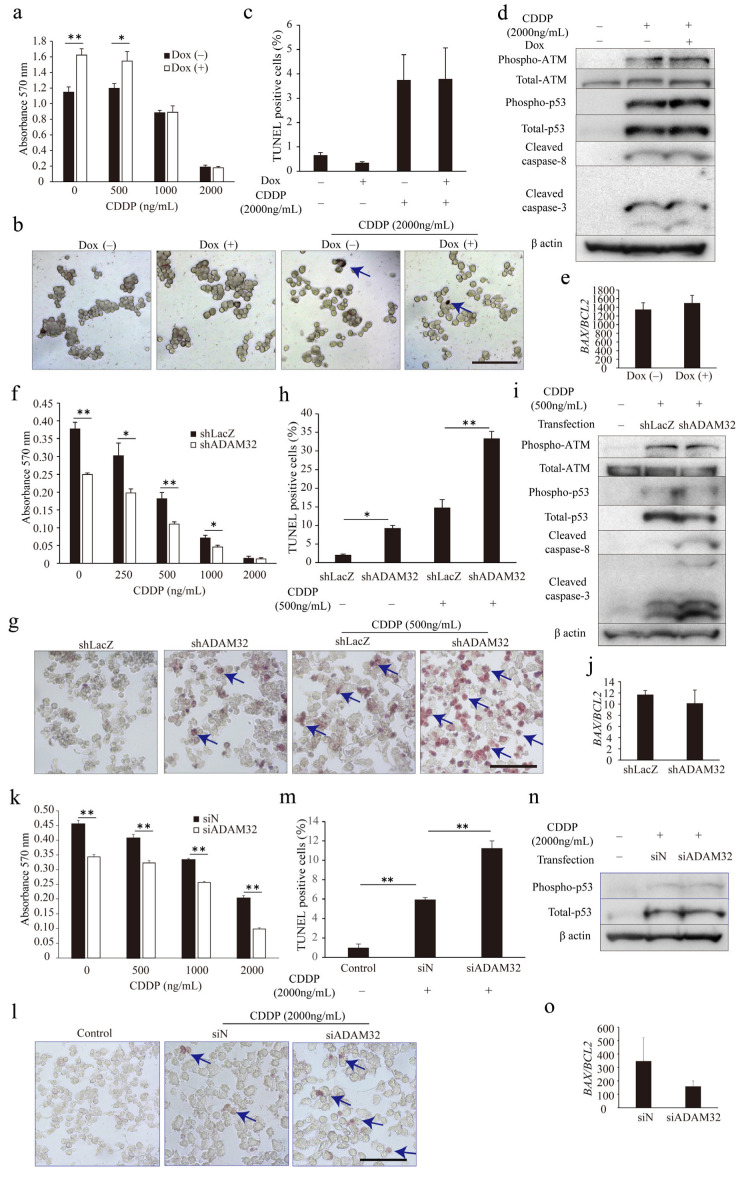
ADAM32 shows antiapoptotic function in CDDP-treated cancer cells. (**a**) HepG2 Tet cells were incubated with or without Dox for 96 h and then treated with CDDP at different concentrations, as indicated, for 72 h. Cell viability in each group was evaluated with the MTT assay; average of each sample is shown (n = 4). (**b**) Representative images are shown from the TUNEL assay using HepG2 Tet cells. (**c**) TUNEL-stained cells were counted in five independent fields, and the percentage of TUNEL-positive cells was calculated (n = 3). (**d**) HepG2 Tet cells were incubated with or without Dox for 96 h and treated with 2000 ng/mL of CDDP for 48 h. The expression levels of apoptosis-related proteins were evaluated by immunoblotting. (**e**) Expression levels of *BAX* and *BCL2* in HepG2 Tet cells were evaluated with real-time RT-PCR. *BAX*/*BCL2* ratios are shown (n = 3). (**f**) HepG2 or (**k**) HBCs were transiently transfected with shRNA or siRNA for 24 h and then treated with CDDP at various concentrations for 48 h or 72 h. Cell viability in each group was evaluated with the MTT assay (n = 4). Representative images are shown from the TUNEL assay using (**g**) HepG2 Tet or (**l**) HBCs. The percentages of TUNEL-positive cells are shown for (**h**) HepG2 (n = 3) or (**m**) HBCs (n = 4). (**i**) HepG2 or (**n**) HBCs were transiently transfected with shRNA or siRNA for 24 h and then treated with 500 or 2000 ng/mL of CDDP for 48 h. Expression levels of apoptosis-related proteins were evaluated by immunoblotting. *BAX*/*BCL2* ratios in HepG2 or HBCs are shown for (**j**) HepG2 (n = 3) or (**o**) HBCs (n = 3). Values are summarized by mean and SE. * *p* < 0.05; ** *p* < 0.01. (**b**,**g**,**l**) Blue arrows indicate TUNEL-positive cells. (**d**,**i**,**n**) Representative images are shown from three independent experiments. Scale bar = 100 μm. All the whole immunoblotting figures can be found in Appendix A.

**Figure 6 cancers-14-04732-f006:**
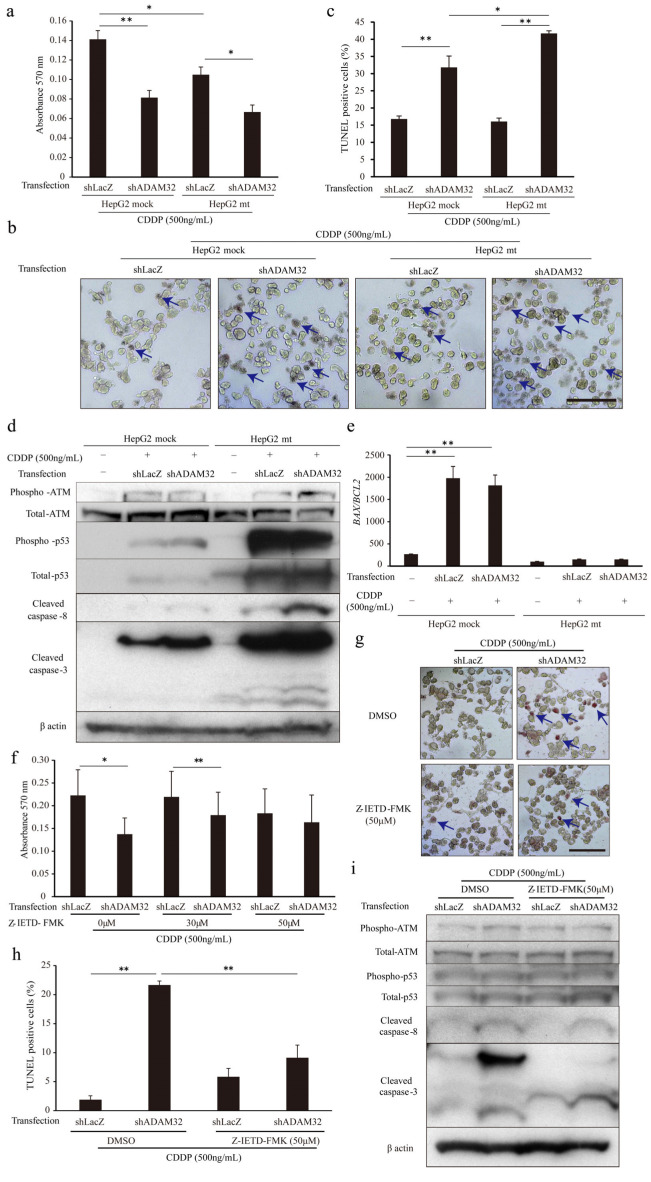
ADAM32 regulates antiapoptotic function through caspase-8-related signals. (**a**) Cell viability of HepG2 mock and HepG2 mt cells transiently transfected with shLacZ or shADAM32 and treated with 500 ng/mL CDDP evaluated with the MTT assay. The average of each sample is shown (n = 5). (**b**) Representative images from the TUNEL assay using HepG2 mock and HepG2 mt under different conditions. (**c**) Percentages of TUNEL-positive cells were calculated, and the average of each sample is shown (n = 3). (**d**) HepG2 mock and HepG2 mt cells transfected with or without shRNA vectors and treated with or without 500 ng/mL CDDP for 48 h. Expression levels of apoptosis-related proteins were evaluated by immunoblotting. (**e**) Expression levels of *BAX* and *BCL2* in HepG2 mock or HepG2 mt cells evaluated by real-time RT-PCR. *BAX*/*BCL2* ratios are shown (n = 3). (**f**) HepG2 cells treated with DMSO or 30 or 50 μM Z-IETD-FMK for 1 h and then transfected with shRNA vectors. After 24 h, cells were treated with 500 ng/mL of CDDP for 72 h. Cell viability in each group was evaluated in the MTT assay. The average of five samples is shown. (**g**) Representative images from the TUNEL assay using HepG2 treated with DMSO or Z-IETD-FMK similarly to (**f**) are shown. (**h**) Percentages of TUNEL-positive cells (n = 3). (**i**) Expression levels of apoptosis-related proteins evaluated by immunoblotting. Values are summarized as mean and SE. * *p* < 0.05; ** *p* < 0.01. (**b**,**d**,**g**,**i**) Representative images from three independent experiments are shown. (**b**,**g**) Blue arrows indicate TUNEL-positive cells. Scale bar = 100 μm. All the whole immunoblotting figures can be found in Appendix A.

**Table 1 cancers-14-04732-t001:** Expression levels of *ADAMs* and *ADAMTSs* in the tumor and NCL of patients with HBL.

Gene Symbol	Official Full Name	Fold Change	adj.*p*.Val ^a^	*p*-Value ^b^
Upregulated (>1.5 fold)				
*ADAM32*	ADAM metallopeptidase domain 32	2.10243	4.57 × 10^−6^	7.86 × 10^−8^
*ADAMTS6*	ADAM metallopeptidase with thrombospondin type 1 motif 6	1.85419	2.06 × 10^−4^	1.11 × 10^−5^
*ADAM9*	ADAM metallopeptidase domain 9	1.82631	1.77 × 10^−3^	1.76 × 10^−4^
Downregulated (>1.5 fold)				
*ADAMTS1*	ADAM metallopeptidase with thrombospondin type 1 motif	−2.16599	1.95 × 10^−2^	3.78 × 10^−3^
*ADAM19*	ADAM metallopeptidase domain 19	−1.87116	2.08 × 10^−3^	2.16 × 10^−4^
*ADAMTS13*	ADAM metallopeptidase with thrombospondin type 1 motif 13	−1.75295	3.57 × 10^−7^	2.99 × 10^−9^
*ADAMTS17*	ADAM metallopeptidase with thrombospondin type 1 motif 17	−1.51324	3.59 × 10^−4^	2.25 × 10^−5^

^a^ *p* values after multiple-testing correction. ^b^ Raw *p* values.

## Data Availability

The data that support the findings of this study are openly available in Gene Expression Omnibus as GSE131329.

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
