# Peer review of "Oncogenic Role of ADAM32 in Hepatoblastoma: A Potential Molecular Target for Therapy"

_cancers, 2022, doi:10.3390/cancers14194732_

Round 1

Reviewer 1 Report (Previous Reviewer 2)

All of the questions posed have been meticulously answered by the authors.  The article can be accepted for the publication. 

Author Response

Thank you very much for your careful and supportive reviewing.

Reviewer 2 Report (New Reviewer)

The authors did good revisions accordingly but maybe better to address the following points further.

1. For the Kaplan-Meier survival assays, it’s better to also include HCC to show the expression levels of ADAM32 in HCC even though it might not be related to prognosis of HCC patients. It’s better to show all the data not just the significant ones particularly HBL also is one of the liver cancers. Also, for the Kaplan-Meier survival assays, it is important to show the sample number (N) in the output results.

Are the ADAM32 expression data used here in different cancers from public database?

2. line 257-258, for the Fig. S4, the authors analyzed ADAM32/9 relative mRNA expression levels across different tissues. Are the data from NCBI database or other database like human protein atlas? it's better to indicate the source.

Author Response

  1. For the Kaplan-Meier survival assays, it’s better to also include HCC to show the expression levels of ADAM32 in HCC even though it might not be related to prognosis of HCC patients. It’s better to show all the data not just the significant ones particularly HBL also is one of the liver cancers. Also, for the Kaplan-Meier survival assays, it is important to show the sample number (N) in the output results.

Are the ADAM32 expression data used here in different cancers from public database?

Response 1: Thank you very much for your helpful suggestion. It actually improved our study. To evaluate prognostic value of ADMA32 expression in various cancers, we performed Kaplan-Meier survival assay using web-based tool for survival analysis, Kaplan-Meier Plotter (http://kmplot.com/analysis). Since available data sets of microarray in the tool were limited to breast, ovarian, lung, and gastric cancers at the original analyses, we showed the results of those cancers. Following your suggestion, we have tried to analyze in HCC, and found data sets of RNA-seq and added the results of HCC in figure S5f.

Sample number of each analysis was shown in the figures.

The data sets used in the web tool were obtained from GEO, EGA, and TCGA as shown in Kaplan-Meier Plotter web site and related articles (http://kmplot.com/analysis and J Med Internet Res. 2021 Jul 26;23(7):e27633. doi: 10.2196/27633.). We have revised the related figure and sentences (please see Figure S5 and page 6, lines 268-269; Supplementary Materials page 5, lines 49-52; and page 12, lines 216-218).

  1. line 257-258, for the Fig. S4, the authors analyzed ADAM32/9 relative mRNA expression levels across different tissues. Are the data from NCBI database or other database like human protein atlas? it's better to indicate the source.

Response 2: We compared the expression levels of ADAM32 by cap analysis gene expression (CAGE), which are obtained from the Functional Annotation of the Mammalian Genome project (FANTOM) or RefEx (https://refex.dbcls.jp/index.php?lang=en). We have revised the related sentences in the Materials and Methods to add detail sources of the data (please see page 2, lines 90-92; and).

This manuscript is a resubmission of an earlier submission. The following is a list of the peer review reports and author responses from that submission.

Round 1

Reviewer 1 Report

The authors tried to report that expression level of ADAM32 is particularly high in hepatoblastoma  tissue samples and is associated with poor prognosis in various cancer types through  the regulation of cancer cell proliferation, stemness, migration, invasion, and acquired resistance to chemo-therapy.

There are several major concerns: 1. In Materials and Methods, they said they did DNA microarray analysis. Actually, the dataset they used is a RNA dataset. 2. The author concluded that the expression level of ADAM32 is particularly high in hepatoblastoma tissue, however, reviewer found that ADAM32  is not even in the list of 250 top differentially  expressed genes in this dataset. 3. They used HepG2 cell line, a human hepatocellular carcinoma cell line, not a true hepatoblastoma cell line and a breast cancer cell line MCF7.  For hepatoblastoma study, at least 2 hepatoblastoma cell lines should be used.  Using a breast cancer cell line in the study is weird. 4. Using TCGA data, reviewer found the level of ADAM32 is not related to survival of hepatocellular carcinoma, the most related one. 5. There is no data  (either immunostaining or western blotting) to support that ADAM32 was expressed/increased in human  hepatoblastoma tumor samples. Due to these concerns, the validity and significance  of  the forced expression or knockdown experiments are questionable.   

Reviewer 2 Report

Hepatoblastoma is the most common malignant liver tumor in children (HBL). In the last decade, several important genetic or molecular events that appear to be important for the development and progression of hepatoblastoma have been identified. Better knowledge of the molecular alterations that underpin hepatoblastoma could lead to improved patient outcomes. ADAMs (a disintegrin and metalloprotease) have been found in high levels in a variety of human carcinomas, including hepatoblastoma, and are thought to play a key role in cancer progression. The authors of this paper revealed ADAM32 biological roles in HBL and its potential as an anticancer therapeutic target. However, I have a few concerns about this research.

1. In figures 4 and 5, the majority of protein expression is of poor quality, with some protein bands exhibiting upwards and others displaying downwards faces, smiley faces, and so on. Some of the bands appear to be non-specific in the supplementary figures. Furthermore, no protein ladder was visible on any of the blots. The author did not appear to test three times. This data is extremely bad in general.

2. The authors demonstrated that ADAM32 enhances the expression levels of stem cell and EMT-associated genes in figure 2 and figure 3 using a real-time RT-PCR graph. The author should demonstrate some key genes involved in protein expression. They should also examine the E-, P-, and N-cadherins.

3. The authors should test the biological role of ADAM32 in at least two HBL cell lines as well as a control cell line. This manuscript does not permit the use of MCF7 breast cancer cells data.

4. To see the outcomes, an in vivo experiment and IHC should be carried out.